# DebGCD: Debiased Learning with Distribution Guidance for Generalized Category Discovery

**Yuanpei Liu**      **Kai Han**[*]

Visual AI Lab, The University of Hong Kong

`ypliu0@connect.hku.hk`      `kaihanx@hku.hk`

## ABSTRACT

In this paper, we tackle the problem of Generalized Category Discovery (GCD). Given a dataset containing both labelled and unlabelled images, the objective is to categorize all images in the unlabelled subset, irrespective of whether they are from known or unknown classes. In GCD, an inherent label bias exists between known and unknown classes due to the lack of ground-truth labels for the latter. State-of-the-art methods in GCD leverage parametric classifiers trained through self-distillation with soft labels, leaving the bias issue unattended. Besides, they treat all unlabelled samples uniformly, neglecting variations in certainty levels and resulting in suboptimal learning. Moreover, the explicit identification of semantic distribution shifts between known and unknown classes, a vital aspect for effective GCD, has been neglected. To address these challenges, we introduce DebGCD, a Debiased learning with distribution guidance framework for GCD. Initially, DebGCD co-trains an auxiliary debiased classifier in the same feature space as the GCD classifier, progressively enhancing the GCD features. Moreover, we introduce a semantic distribution detector in a separate feature space to implicitly boost the learning efficacy of GCD. Additionally, we employ a curriculum learning strategy based on semantic distribution certainty to steer the debiased learning at an optimized pace. Thorough evaluations on GCD benchmarks demonstrate the consistent state-of-the-art performance of our framework, highlighting its superiority. Project page: https://visual-ai.github.io/debgcd/

## 1 INTRODUCTION

Over the years, the field of computer vision has witnessed remarkable progress in diverse tasks such as object detection Girshick (2015); Ren et al. (2015), classification Simonyan & Zisserman (2015); He et al. (2016), and segmentation He et al. (2017); Wang et al. (2020). These advancements have predominantly stemmed from the availability of expansive labelled datasets Deng et al. (2009); Lin et al. (2014). However, the prevalent insufficiency of training data in real-world scenarios is a noteworthy concern. This has engendered a surge in research on semi-supervised learning Chapelle et al. (2009) and self-supervised learning Jing & Tian (2020), yielding promising outcomes in comparison to supervised learning approaches. Recently, the task of category discovery, which was initially studied as novel category discovery (NCD) Han et al. (2019) and subsequently extended to its relaxed variant, generalized category discovery (GCD) Vaze et al. (2022b), has emerged as a research task attracting increasing attention. GCD considers a partially-labelled dataset, where the unlabelled subset may contain instances from both labelled and unseen classes. The objective is to learn to transfer knowledge from labelled data to categorize unlabelled data.

In GCD, there exists an inherent label bias between known and unknown classes due to the absence of ground-truth labels for the latter. This label bias has the potential to cause the model to inadvertently develop a decision rule making confident predictions that inclined to known classes. Similar problem has been identified in the area of long-tailed recognition Tang et al. (2020); Yang et al. (2022). Besides, in other fields such as object classification Choi et al. (2019); Bahng et al. (2020); Geirhos et al. (2020), it is widely known that model performance suffers from task-specific bias.

---

[*]Corresponding author.

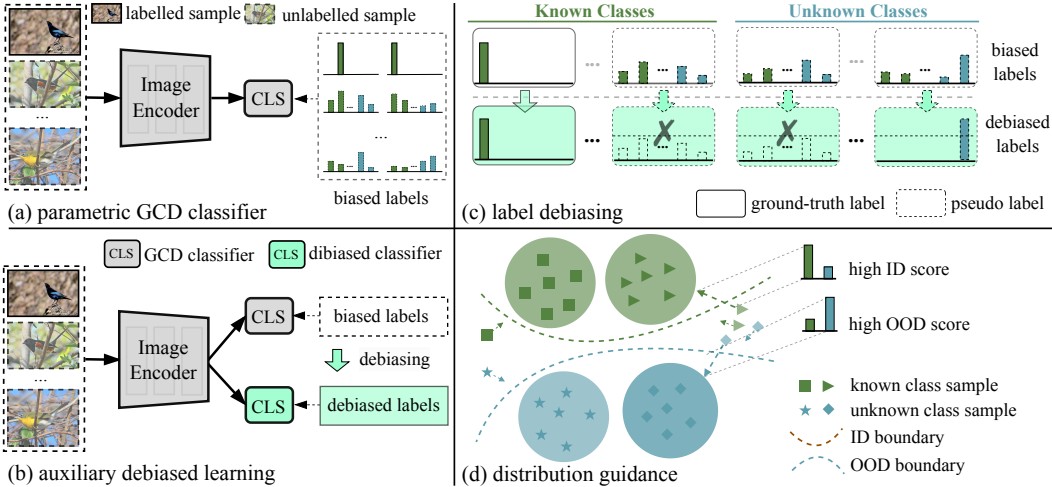

Figure 1: (a) The parametric GCD classifier Wen et al. (2023) is trained on labelled and unlabelled images using ground-truth hard labels and soft labels, respectively. (b) The auxiliary debiased learning: training another classifier using debiased labels. (c) The process of label debiasing: keep the hard labels unchanged and transform soft labels to one-hot hard labels; samples that do not meet the threshold are removed. (d) The illustration of distribution guidance: if a sample receives a high in-distribution/out-of-distribution score, its weight in GCD training will be increased accordingly.

State-of-the-art parametric classifier methods in GCD, such as those proposed by Wen et al. (2023); Zhao et al. (2023); Vaze et al. (2023), leverage the self-distillation Caron et al. (2021) mechanism based on soft labels generated from the model's predictions of another image view. While these methods have shown promising results, they still rely on biased labels for training (as shown in Fig. 1 (a)). The issue of label bias remains an unattended problem in the realm of GCD. Additionally, existing approaches uniformly handle all unlabelled samples without explicitly accounting for their different certainty, which may introduce noise to the model training due to unreliable samples. Moreover, they do not explicitly address semantic shifts, especially in a scenario like GCD involving both known and unknown classes within unlabelled data. Notably, these concerns have been demonstrated to provide significant advantages in related tasks, such as open-world semi-supervised learning Cao et al. (2022). In this area, OpenCon Sun & Li (2022) has attempted to identify novel samples based on their proximity to known prototypes. However, its performance is heavily contingent on predefined distance thresholds, ultimately yielding suboptimal accuracy.

We propose DebGCD, a novel framework designed to tackle the challenges of GCD. DebGCD introduces Debiased learning with distribution guidance for GCD, incorporating several innovative techniques specifically tailored for this task. Firstly, we introduce a novel auxiliary debiased learning paradigm for GCD (as shown in Fig. 1(b) and (c)). This method entails training an auxiliary debiased classifier in the same feature space as the GCD classifier. Unlike the GCD classifier, both labelled and unlabelled data are trained using one-hot hard labels to prevent label bias between known and unknown classes. Secondly, to discern the semantic distribution of unlabelled samples, we propose to learn a semantic distribution detector in a decoupled normalized feature space, which we empirically find it enhance the learning effect of GCD implicitly. Furthermore, we propose to measure the certainty of a sample based on its semantic distribution detection score. This certainty score then enables the gradual inclusion of unlabelled samples from both known and unknown classes during training, allowing the auxiliary debiased learning to function in a curriculum learning approach (as shown in Fig. 1(d)), thus further enhancing its performance. We develop our framework upon the parametric baseline Wen et al. (2023). By effectively incorporating these components into a unified framework, DebGCD can be trained end-to-end in a single stage while not introducing any additional computational burden during inference. Despite its simplicity, DebGCD attains unparalleled performance on the public GCD datasets, including the generic classification datasets CIFAR-10 Krizhevsky et al. (2009), CIFAR-100 Krizhevsky et al. (2009), and ImageNet Deng et al. (2009), as well as the fine-grained SSB Vaze et al. (2022a) benchmark.

We make the following key contributions in this work: (1) We propose DebGCD, a novel framework that addresses the challenging GCD task by considering both label bias and semantic shift, marking

the first exploration of these aspects for the challenging GCD task. (2) Within DebGCD, we propose a novel auxiliary debiased learning paradigm to optimize the clustering feature space, in conjunction with the distribution shift detector in a distinct feature space. They work tightly to enhance the model's discovery capabilities. (3) We introduce a curriculum learning mechanism that steers the debiased learning process using a distribution certainty score, effectively mitigating the negative impact of uncertain samples. (4) Through extensive experimentation on public GCD benchmarks, DebGCD consistently demonstrates its effectiveness and achieves superior performance.

## 2 RELATED WORK

**Category Discovery.** This task is initially studied as novel category discovery (NCD) Han et al. (2019), aiming to discover categories from unlabelled data consisting of samples from novel categories, by transferring the knowledge from the labelled categories. Many methods have been proposed to tackle NCD, such as Han et al. (2019; 2020; 2021); Fini et al. (2021); Zhao & Han (2021); Joseph et al. (2022). Vaze et al. (2022b) extends NCD to a more relaxed task, generalized category discovery (GCD), wherein unlabelled datasets encompass both known and unknown categories. A baseline method is presented for this task, incorporating self-supervised representation learning and semi-supervised $k$-means clustering, and extending popular NCD methods such as RankStats Han et al. (2020) and UNO Fini et al. (2021) to GCD. CiPR Hao et al. (2024) proposes to bootstrap the representation by leveraging cross-instance positive relations in the partially labelled data for contrastive learning. Cao et al. (2022) addresses a similar problem to GCD from the perspective of semi-supervised learning. SimGCD Wen et al. (2023) introduces a strong parametric baseline achieving promising performance improvements. In Vaze et al. (2023), a new dataset is introduced to illustrate the limitations of unsupervised clustering in GCD. To address these limitations, a method based on the 'mean-teachers' approach is proposed. In Rastegar et al. (2023), a category coding approach is introduced, considering category prediction as the outcome of an optimization problem. Recently, SPTNet Wang et al. (2024b) is proposed to consider the spatial property of images and presents a spatial prompt tuning method, enabling the model to better focus on object parts for knowledge transfer. Moreover, an increasing number of efforts are focused on addressing category discovery from diverse perspectives. For example, Jia et al. (2021) concentrates on multi-modal category discovery, whereas Zhang et al. (2022), Ma et al. (2024), and Cendra et al. (2024) investigate continual category discovery. Additionally, Pu et al. (2024) explores federated category discovery. Furthermore, Wang et al. (2025) studies category discovery in the presence of domain shifts.

**Debiased Learning.** The issue of bias in data and the susceptibility of machine learning algorithms to such bias have been widely recognized as crucial challenges across diverse tasks. Numerous methodologies have been developed to address and alleviate biases inherent in training datasets or tasks. The studies by Ponce (2006); Torralba & Efros (2011) elucidate that many training sets impose regularity conditions that are impractical in real-world settings, leading to machine learning models trained on such data failing to generalize in the absence of these conditions. Furthermore, recent research by Hendrycks et al. (2021); Xiao et al. (2021); Li et al. (2021) demonstrate biases in state-of-the-art object recognition models towards specific backgrounds or textures associated with object classes. Additionally, Sagawa et al. (2020) investigate the vulnerability of overparametrized models to spurious correlations, resulting in elevated test errors for minority groups. Notably, large language models also exhibit biased predictions towards certain genders or races, as indicated by Cheng et al. (2021). Furthermore, the severity of biased predictions and fairness concerns related to deployed models are extensively explored across various tasks Zemel et al. (2013); Noble (2018); Bolukbasi et al. (2016). In this paper, we examine the inherent *label bias* in GCD, representing the initial exploration of this issue.

**Out-of-distribution Detection.** In the realm of out-of-distribution (OOD) detection, the objective is to identify samples or data points that originate from a distribution distinct from the one on which the model was trained, encompassing both semantic and domain distributions Yang et al. (2021); Wang et al. (2024a). The simplest method in this area involves utilizing the predicted softmax class probability to detect OOD samples Hendrycks & Gimpel (2017). ODIN Liang et al. (2018) further enhances this approach by introducing temperature scaling and input pre-processing. Additionally, Bendale & Boult (2016) proposes an alternative approach by calculating the score for an unknown class using a weighted average of all other classes. OOD detection has been applied in various open-set tasks, such as open-set semi-supervised learning Yu et al. (2020) and universal domain adaptation Saito & Saenko (2021), where it is utilized to select in-distribution data during training.

In contrast, our focus lies in the exploration of semantic shift detection considering the specific challenges of GCD. OpenCon Sun & Li (2022) has attempted to explore the semantic shift for open-world semi-supervised learning. However, its reliance on a predefined distance threshold to rigidly distinguish inliers and outliers leads to suboptimal accuracy. In contrast, our method takes a distinct approach by avoiding a rigid separation. We subtly utilize the predicted OOD score by our model as a guiding factor for debiased learning, further enabling a curriculum learning scheme.

## 3 PRELIMINARIES

### 3.1 PROBLEM STATEMENT

Generalized category discovery (GCD) aims to learn a model that can not only correctly classify the unlabelled samples of known categories but also cluster those of unknown categories. Given an unlabelled dataset $\mathcal{D}_u = \{(\boldsymbol{x}_i^u, y_i^u)\} \in \mathcal{X} \times \mathcal{Y}_u$ and a labelled dataset $\mathcal{D}_l = \{(\boldsymbol{x}_i^l, y_i^l)\} \in \mathcal{X} \times \mathcal{Y}_l$, where $\mathcal{Y}_u$ and $\mathcal{Y}_l$ are their label sets respectively. The unlabelled dataset contains samples from both known and unknown categories, *i.e.*, $\mathcal{Y}_l \subset \mathcal{Y}_u$. The number of labelled categories is $M = |\mathcal{Y}_l|$. We assume the number of categories $K = |\mathcal{Y}_l \cup \mathcal{Y}_u|$ to be known following previous works Han et al. (2021); Wen et al. (2023); Vaze et al. (2023). When it is unknown, methods like Han et al. (2019); Vaze et al. (2022b) can be applied to provide a reliable estimation.

### 3.2 BASELINE

Wen et al. (2023) introduces a robust parametric GCD baseline, which has been widely adopted in the field ever since Vaze et al. (2023); Wang et al. (2024b). It employs a parametric classifier, implemented in a self-distillation manner Caron et al. (2021). The classifier is randomly initialized with $K$ normalized category prototypes $\mathcal{C} = \{\boldsymbol{c}_1, ..., \boldsymbol{c}_K\}$. For the randomly augmented view of an image $\boldsymbol{x}_i$ and its corresponding normalized hidden feature vector $\boldsymbol{h}_i = \phi(\boldsymbol{x}_i)/||\phi(\boldsymbol{x}_i)||$, the output probability for the $k$th category is given by:

$$\boldsymbol{p}_i^{(k)} = \frac{\exp(\boldsymbol{h}_i \cdot \boldsymbol{c}_k / \tau_s)}{\sum_{j=1}^{K} \exp(\boldsymbol{h}_i \cdot \boldsymbol{c}_j / \tau_s)}, \tag{1}$$

where $\tau_s$ is the scaling temperature for this *'student'* view. The soft label $\boldsymbol{q}_i$ is produced by the *'teacher'* view with a sharper temperature $\tau_t$ using another augmented view in the same fashion. The self-distillation loss of the two views is then simply calculated following the cross-entropy loss $\ell_{ce}(\boldsymbol{q}', \boldsymbol{p}) = -\sum_{j=1}^{K} \boldsymbol{q}'^{(j)} \log \boldsymbol{p}^{(j)}$. Given a mini-batch $\mathcal{B}$ containing both labelled samples $\mathcal{B}_l$ and unlabelled images $\mathcal{B}_u$, the self-distillation loss is calculated using all the samples in the mini-batch:

$$\mathcal{L}_{cls}^u = \frac{1}{|\mathcal{B}|} \sum_{i \in \mathcal{B}} \ell_{ce}(\boldsymbol{q}_i', \boldsymbol{p}_i) - \xi H(\overline{\boldsymbol{p}}), \tag{2}$$

where $\overline{\boldsymbol{p}} = \frac{1}{2|\mathcal{B}|} \sum_{i \in \mathcal{B}} (\boldsymbol{p}_i + \boldsymbol{p}_i')$ denotes the mean prediction within a batch and its entropy $H(\overline{\boldsymbol{p}}) = -\sum_{j=1}^{K} \overline{\boldsymbol{p}}^{(j)} \log \overline{\boldsymbol{p}}^{(j)}$ weighted by $\xi$. For the labelled samples, the supervised classification loss is written as $\mathcal{L}_{cls}^s = \frac{1}{|\mathcal{B}_l|} \sum_{i \in \mathcal{B}_l} \ell_{ce}(\boldsymbol{p}_i, \boldsymbol{y}_i)$, where $\boldsymbol{y}_i$ represents the one-hot vector corresponding to the ground-truth label $y_i$. The whole classification objective is $\mathcal{L}_{cls} = (1 - \lambda_b^{gcd})\mathcal{L}_{cls}^u + \lambda_b^{gcd}\mathcal{L}_{cls}^s$. Combining with the representation learning loss $\mathcal{L}_{rep}$ adopted from Vaze et al. (2022b), the overall training objective becomes:

$$\mathcal{L}_{gcd} = \mathcal{L}_{cls} + \mathcal{L}_{rep}. \tag{3}$$

Through training with $\mathcal{L}_{gcd}$ on both $\mathcal{D}_l$ and $\mathcal{D}_u$, the classifier can directly predict the labels for unlabelled samples after training.

## 4 DEBIASED LEARNING WITH DISTRIBUTION-GUIDANCE FOR GCD

In this section, we present our debiased Learning with distribution-guidance framework for GCD (see Fig. 2). First, in Sec. 4.1, we present the semantic distribution learning on the GCD task. Next, in Sec. 4.2, we demonstrate the training paradigm of the debiased classifier. Finally, we describe the joint training and inference process of our full framework in Sec. 4.3.

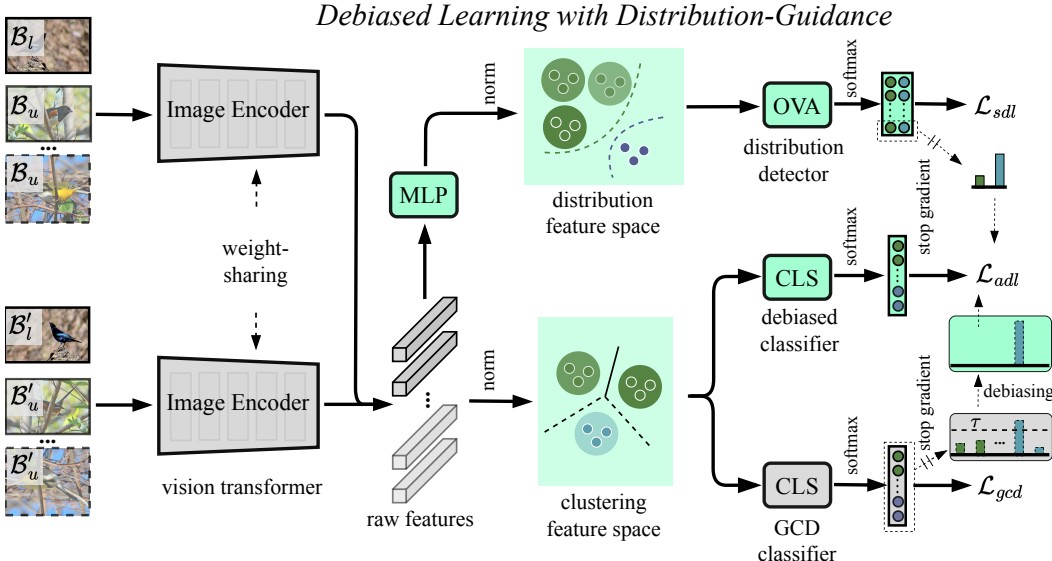

Figure 2: Overview of the DebGCD framework. In the upper branch, raw features are transformed using an MLP and then normalized. These normalized features are used for semantic distribution learning with a one-vs-all classifier. In the lower branch, a GCD classifier is trained on the normalized raw features. The predictions from both branches are combined to train the debiased classifier. As DebGCD aligns with prior work in representation learning, it's not explicitly depicted here.

## 4.1 LEARNING SEMANTIC DISTRIBUTION

OOD detection methods have been employed in tasks like universal domain adaptation Saito & Saenko (2021) and open-set semi-supervised learning Yu et al. (2020), obtaining improved performance. In these tasks, the identified OOD samples are treated as a single *background* class to avoid affecting the recognition of unlabelled samples from the labelled classes, and the distribution shifts can be of any type. In GCD, we are particularly interested in identifying the semantic shifts. The instances from the labelled classes are considered in-distribution (ID) samples, while the instances from the novel classes are considered OOD samples. However, the potential of effectively introducing OOD techniques for GCD remains under-explored. An intuitive approach for OOD detection is to examine the class prediction probabilities. Generally, the maximum softmax or logit score of a closed-set classifier can serve as a good indicator of OOD Vaze et al. (2022a); Wang et al. (2024a). However, this is not suitable for the common GCD classifier, which contains an mean entropy regularization term in the loss function to prevent biased predictions. Nevertheless, we find that it also results in the classifier's predictions on known categories being less confident, thereby degrading the OOD detection performance. Moreover, these OOD methods need to carefully select a threshold Geng et al. (2020) for rejecting "unknown" samples, which relies on validation or a pre-defined ratio of "unknown" samples, making them impractical for the GCD due to the absence of such validation samples. One-vs-all (OVA) classifier Saito & Saenko (2021), which has consistently shown promise in the literature Saito et al. (2021); Fan et al. (2023); Li et al. (2023), can be a more suitable option. Moreover, in the context of OOD, the objective is not to differentiate between multiple distinct unknown categories, as we do in GCD; rather, we aim to distinguish all unknown classes from the known classes, effectively framing this as a binary classification problem. This calls for the need of a different feature space that is better suited for this task. Therefore, as depicted in Fig. 2, we introduce an additional multi-layer perceptron (MLP) projection network $\rho_s$, to project raw features into another embedding space, followed by $\ell_2$-normalization to attain the embedding space for distribution discrimination. Different from the prior works applying OOD in the magnitude-aware feature space for other tasks Yu et al. (2020); Saito et al. (2021); Li et al. (2023), we empirically found that the $\ell_2$-normalized feature space aligns more seamlessly with the DINO pre-trained weights in GCD. Subsequently, we devise $M$ $\ell_2$-normalized binary classifiers, denoted as $\chi = \{\chi_1, \chi_2, ..., \chi_M\}$, for semantic OOD detection in GCD.

Given an augmented image view $x_i$, its $\ell_2$-normalized feature in the semantic distribution feature space is $f_i = \rho_s(\phi(x_i))/||\rho_s(\phi(x_i))||$. Subsequently, the output of the $k$-th binary classifier is

$o_{i,k} = \mathrm{softmax}(\chi_k(\boldsymbol{f}_i))$, where $\boldsymbol{o}_{i,k} = (o_{i,k}^+, o_{i,k}^-)$ and $o_{i,k}^+ + o_{i,k}^- = 1$. For labelled samples, a multi-binary cross-entropy loss with a hard-negative sampling strategy Saito et al. (2021) is employed:

$$\mathcal{L}_{sdl}^s = \frac{1}{|\mathcal{B}_l|} \sum_{i \in \mathcal{B}_l} (-\log(o_{i,y_i}^+) - \min_{k \neq y_i} \log(o_{i,k}^-)), \tag{4}$$

where $y_i$ represents the ground-truth category label of the sample $\boldsymbol{x}_i$. For unlabelled samples, an entropy minimization technique Saito & Saenko (2021) is applied to improve low-density separation:

$$\mathcal{L}_{sdl}^u = -\frac{1}{\mathcal{B}_u} \sum_{i \in \mathcal{B}_u} \sum_{j=1}^{M} (o_{i,j}^+ \log(o_{i,j}^+) + o_{i,k}^- \log(o_{i,k}^-)), \tag{5}$$

where $\mathcal{B}_u$ denotes the unlabelled subset in current mini-batch. The loss function for the semantic distribution learning is defined as:

$$\mathcal{L}_{sdl} = \mathcal{L}_{sdl}^s + \mathcal{L}_{sdl}^u. \tag{6}$$

By optimizing $\mathcal{L}_{sdl}$, our detector distinctly segregates the feature distributions between known and unknown categories. Additionally, it generates a predicted score based on the maximum output from all $M$ binary classifiers, denoted as:

$$s_i = o_{i,y_p}^-, y_p = \arg\max_j o_{i,j}^+. \tag{7}$$

This score will serve as a crucial cue for the debiased learning to be introduced next.

## 4.2 AUXILIARY DEBIASED LEARNING

As depicted in Fig. 2, the raw features are normalized to the clustering feature space in the lower branch, wherein novel categories are discovered. In order to minimize the unintended negative impact of biased labels while maintaining the basic probability constraints Assran et al. (2022) and consistency regularization Caron et al. (2021) in the GCD classifier, we propose an auxiliary debiased learning mechanism. Specifically, a parallel debiased classifier $\psi_s$ initialized with $K$ normalized prototypes $\mathcal{C}^a = \{\boldsymbol{c}_1^a, ..., \boldsymbol{c}_K^a\}$, is trained in the same embedding space using debiased labels. Note that in our experiment, we only finetune the last two transformer blocks of the DINO Caron et al. (2021) pre-trained ViT backbone. The $k$-th softmax score of sample $\boldsymbol{x}_i$ is given by:

$$\boldsymbol{p}_i^{a(k)} = \frac{\exp(\boldsymbol{h}_i \cdot \boldsymbol{c}_k^a / \tau_a)}{\sum_{j=1}^{K} \exp(\boldsymbol{h}_i \cdot \boldsymbol{c}_j^a / \tau_a)}, \tag{8}$$

where $\tau_a$ is the scaling temperature. The maximum classification score has demonstrated promising performance in several semi-supervised learning methods and we find it also a good indicator of sample quality in the context of GCD task. For an augmented view $\boldsymbol{x}_i$ and its GCD classifier prediction $\boldsymbol{p}_i$, a debiasing threshold $\tau$ is set on the $\max(\boldsymbol{p}_i)$, with only samples surpassing $\tau$ being utilized to train the debiased classifier, expressed as $\mathbb{1}(\max(\boldsymbol{p}_i) > \tau)$. Additionally, given that the semantic distribution detector and the GCD classifier are learned in different feature spaces and paradigms, it is essential to ensure the alignment of their predictions. Consequently, we introduce a function to indicate the task consistency of these two tasks, defined as:

$$\mathcal{F}(\hat{y}_i, s_i) = \mathbb{1}(\hat{y}_i \in \mathcal{Y}_u \wedge s_i > 0.5) \vee \mathbb{1}(\hat{y}_i \in \mathcal{Y}_l \wedge s_i < 0.5), \tag{9}$$

where $\hat{y}_i = \arg\max(\boldsymbol{p}_i)$ represents the predicted category index by the GCD classifier, and $\hat{\boldsymbol{y}}_i$ denotes its corresponding one-hot vector. This function aims to selectively filter out samples with identical distribution predictions across the two tasks.

Furthermore, as previously stated, given the inclusion of both known (in-distribution) and unknown (out-of-distribution) samples in the unlabelled data, it is imperative to devise a learning strategy based on semantic distribution information. With the training progresses, the semantic OOD scores gradually approach the two extremes (*i.e.*, 0 and 1). The score of the unknown class sample steadily increases to 1, while the score of the known class gradually decreases to 0. Prior techniques Saito et al. (2021); Li et al. (2023) simply employ a threshold to determine whether the sample belongs to the known or unknown. Such a naïve method is unreliable and may introduce many noises to the model training for GCD. In our approach, we prioritize samples with distinct distributions for

---

**Algorithm 1** End-to-end Training Algorithm for DebGCD.

---

**Input**: Set of labelled data $\mathcal{D}_l = \{(\boldsymbol{x}_i^l, y_i^l)\}$, set of unlabelled data $\mathcal{D}_u = \{(\boldsymbol{x}_i^u, y_i^u)\}$. Data augmentation function $\mathcal{A}$. Model parameters $w$, learning rate $\eta$, epoch $E_{max}$, iteration $I_{max}$, trade-off parameters, $\lambda_{sdl}$, $\lambda_{adl}$;

**for** $Epoch = 1 \ to \ E_{max}$ **do**

    **for** $Iteration = 1 \ to \ I_{max}$ **do**

        **Sample** labelled data $\mathcal{B}_l$, unlabelled data $\mathcal{B}_u$; $i \in \mathcal{B}_u$

        **Compute** model prediction $\boldsymbol{p}_i$, $\boldsymbol{p}_i^a$, $s_i$; loss function $\mathcal{L}_{gcd}$, $\mathcal{L}_{sdl}$           // Eq.3,6,8

        **Compute** debiased label $\hat{y}_i$; task consistency $\mathcal{F}(\hat{y}_i, s_i)$                // Eq.9

        **Compute** loss function $\mathcal{L}_{adl}^s$, $\mathcal{L}_{adl}^u$, $\mathcal{L}_{adl}$                      // Eq.11,12,13

        **Compute** loss function $\mathcal{L}_{all} = \mathcal{L}_{gcd} + \lambda_{sdl}\mathcal{L}_{sdl} + \lambda_{adl}\mathcal{L}_{adl}$

        **Update** model parameters $w = w - \eta \bigtriangledown_w \mathcal{L}_{all}$

    **end**

**end**

**Output**: Model parameter $w$.

---

self-training, aligning with the principles of curriculum learning. To establish a consistent metric for assessing sample discriminability, we introduce a normalized distribution certainty score:

$$d_i = |2 \times s_i - 1|, \tag{10}$$

which approaches the value $0$ for ambiguous samples and the value $1$ for certain samples. This score, to a certain extent, indicates the learning status of samples and can serve as a crucial cue for our debiased classifier. Therefore, the auxiliary debiased learning loss for unlabelled samples is written as:

$$\mathcal{L}_{adl}^u = \frac{1}{\mathcal{B}_u} \sum_{i \in \mathcal{B}_u} \mathbb{1}(\max(\boldsymbol{p_i}) > \tau) \times \mathcal{F}(\hat{y}_i, s_i) \times d_i \times \ell_{ce}(\boldsymbol{p}_i^a, \hat{\boldsymbol{y}}_i). \tag{11}$$

In this manner, the training of the debiased classifier transforms into a curriculum learning process, where easily identifiable samples that are clearly semantic in-distribution or out-of-distribution are given higher priority for learning. Moreover, our debiased classifier also retains the prior knowledge from the labelled data. For the labelled samples, it's is simply trained with the cross-entropy loss:

$$\mathcal{L}_{adl}^s = \frac{1}{\mathcal{B}_l} \sum_{i \in \mathcal{B}_l} \ell_{ce}(\boldsymbol{p}_i^a, \boldsymbol{y}_i). \tag{12}$$

Finally, the overall training loss for the debiased classifier is:

$$\mathcal{L}_{adl} = \mathcal{L}_{adl}^s + \mathcal{L}_{adl}^u. \tag{13}$$

In this manner, all the samples are trained using one-hot hard labels, irrespective of their belongings to known or unknown categories. Operating within the same feature space, our debiased classifier collaborates closely with the GCD classifier, thereby facilitating the joint optimization of the clustering feature space.

## 4.3 LEARNING AND INFERENCE FRAMEWORK

Based on the baseline GCD classifier, our framework is designed to be trained in a multi-task manner. Different from previous approaches in the open-set literature Yu et al. (2020), our DebGCD framework employs a *one-stage* training process, eliminating the necessity for task-specific warm-up phases. Consequently, the three tasks can be jointly trained end-to-end with the overall loss:

$$\mathcal{L}_{all} = \mathcal{L}_{gcd} + \lambda_{sdl}\mathcal{L}_{sdl} + \lambda_{adl}\mathcal{L}_{adl}, \tag{14}$$

where $\lambda_{sdl}$ and $\lambda_{adl}$ denote the loss weights for the semantic distribution detector and debiased classifier, respectively. The complete training pipeline of the framework is illustrated in Algorithm 1.

Throughout the joint training process, the three branches are collectively optimized in an end-to-end manner. During inference, only the GCD classifier is retained. This indicates that our method does not impose any additional computational overhead compared to the baseline approach during inference, further emphasizing its simplicity and efficiency.

## 5 EXPERIMENTS

In this section, we present a comprehensive evaluation of the proposed DebGCD framework and further perform meticulous ablation studies to showcase the effectiveness of its individual components. More results and analysis can be found in the Appendix.

### 5.1 EXPERIMENTAL SETUP

**Datasets.** We conduct a comprehensive evaluation of our method across diverse benchmarks, encompassing the generic image recognition benchmark (CIFAR-10/100 Krizhevsky et al. (2009), ImageNet-100 Deng et al. (2009)), the Semantic Shift Benchmark (SSB) Vaze et al. (2022c) comprising fine-grained datasets CUB Wah et al. (2011), Stanford Cars Krause et al. (2013), and FGVC-Aircraft Maji et al. (2013), along with the challenging ImageNet-1K Deng et al. (2009). For each dataset, we adhere to the data split scheme detailed in Vaze et al. (2022b). The method involves sampling a subset of all classes as the known ('Old') classes $\mathcal{Y}_l$. Subsequently, $50\%$ of the images from these known classes are utilized to construct $\mathcal{D}_l$, while the remaining images are designated as the unlabelled data $\mathcal{D}_u$. The statistics can be seen in Tab. 1.

**Evaluation metrics.** We assess the GCD performance using the clustering accuracy (ACC) in accordance with established conventions Vaze et al. (2022b). For evaluation, the ACC on $\mathcal{D}_l$ is computed as follows, given the ground truth $y_i$ and the predicted labels $\hat{y}_i$:

$$\text{ACC} = \frac{1}{|\mathcal{D}_u|} \sum_{i=1}^{|\mathcal{D}_u|} \mathbb{1}(y_i = h(\hat{y}_i)), \quad (15)$$

where $h$ represents the optimal permutation that aligns the predicted

Table 1: Overview of dataset, including the classes in the labelled and unlabelled sets ($|\mathcal{Y}_l|$, $|\mathcal{Y}_u|$) and counts of images ($|\mathcal{D}_l|$, $|\mathcal{D}_u|$). 'FG' denotes fine-grained.

| Dataset | FG | $|\mathcal{D}_l|$ | $|\mathcal{Y}_l|$ | $|\mathcal{D}_u|$ | $|\mathcal{Y}_u|$ |
|---|---|---|---|---|---|
| CIFAR-10 Krizhevsky et al. (2009) | ✗ | 12.5K | 5 | 37.5K | 10 |
| CIFAR-100 Krizhevsky et al. (2009) | ✗ | 20.0K | 80 | 30.0K | 100 |
| ImageNet-100 Deng et al. (2009) | ✗ | 31.9K | 50 | 95.3K | 100 |
| CUB Wah et al. (2011) | ✓ | 1.5K | 100 | 4.5K | 200 |
| Stanford Cars Krause et al. (2013) | ✓ | 2.0K | 98 | 6.1K | 196 |
| FGVC-Aircraft Maji et al. (2013) | ✓ | 1.7K | 50 | 5.0K | 100 |
| ImageNet-1K Deng et al. (2009) | ✗ | 321K | 500 | 960K | 1000 |

cluster assignments with the ground-truth class labels. ACC for 'All' classes, 'Old' classes and 'New' classes are reported for comprehensive assessment.

**Implementation details.** Following previous attempts in GCD Vaze et al. (2022b); Wen et al. (2023), our model is structured with a ViT-B/16 Dosovitskiy et al. (2021) backbone pre-trained using DINO Caron et al. (2021), and the feature space centers around the 768-dimensional classification token. The projection networks for representation learning and semantic distribution detection comprise three-layer and five-layer MLPs, respectively. The model is trained with a batch size of 128, initiating with an initial learning rate of $10^{-1}$ which decays to $10^{-4}$ using a cosine schedule over 200 epochs. Notably, the loss weights $\lambda_{sdl}$ and $\lambda_{adl}$ are set to 0.01 and 1.0, while the loss balancing weight $\lambda_b^{gcd}$ is assigned to 0.35 following Wen et al. (2023). Regarding the temperature parameters, the initial temperature $\tau_t$ is established at $0.07$, subsequently warmed up to $0.04$ employing a cosine schedule during the first 30 epochs, whereas the other temperatures are set to $0.1$.

### 5.2 BENCHMARK RESULTS

We present benchmark results of our method and compare it with state-of-the-art techniques in generalized category discovery (including ORCA Cao et al. (2022), GCD Vaze et al. (2022b), XCon Fei et al. (2022), OpenCon Sun & Li (2022), PromptCAL Zhang et al. (2023), DCCL Pu et al. (2023), GPC Zhao et al. (2023), CiPR Hao et al. (2024), SimGCD Wen et al. (2023), $\mu$GCD Vaze et al. (2023), InfoSieve Rastegar et al. (2023), and SPTNet Wang et al. (2024b)), as well as robust baselines derived from novel category discovery (RankStats+ Han et al. (2021), UNO+ Fini et al. (2021), and $k$-means MacQueen (1967)). All methods are based on the DINO Caron et al. (2021) pre-trained backbone. This comparative evaluation encompasses performance on the fine-grained SSB benchmark Vaze et al. (2022c) and generic image recognition datasets Krizhevsky et al. (2009); Deng et al. (2009), as shown in Tab. 2 and Tab. 3.

**Results on SSB.** As shown in Tab. 2, DebGCD demonstrates superior performance across the three datasets, achieving an average ACC of $64.4$ on 'All' categories, surpassing the second-best SPT-Net Wang et al. (2024b) by $3\%$. It maintains the best on both Stanford Cars and FGVC-Aircraft

Table 2: Comparison of state-of-the-art GCD methods on SSB Vaze et al. (2022c) benchmark. Results are reported in ACC across the 'All', 'Old' and 'New' categories.

| Method | CUB | | | Stanford Cars | | | FGVC-Aircraft | | | Average |
| --- | --- | --- | --- | --- | --- | --- | --- | --- | --- | --- |
| | All | Old | New | All | Old | New | All | Old | New | All |
| k-means MacQueen (1967) | 34.3 | 38.9 | 32.1 | 12.8 | 10.6 | 13.8 | 16.0 | 14.4 | 16.8 | 21.1 |
| RankStats+ Han et al. (2021) | 33.3 | 51.6 | 24.2 | 28.3 | 61.8 | 12.1 | 26.9 | 36.4 | 22.2 | 29.5 |
| UNO+ Fini et al. (2021) | 35.1 | 49.0 | 28.1 | 35.5 | 70.5 | 18.6 | 40.3 | 56.4 | 32.2 | 37.0 |
| ORCA Cao et al. (2022) | 35.3 | 45.6 | 30.2 | 23.5 | 50.1 | 10.7 | 22.0 | 31.8 | 17.1 | 26.9 |
| GCD Vaze et al. (2022b) | 51.3 | 56.6 | 48.7 | 39.0 | 57.6 | 29.9 | 45.0 | 41.1 | 46.9 | 45.1 |
| XCon Fei et al. (2022) | 52.1 | 54.3 | 51.0 | 40.5 | 58.8 | 31.7 | 47.7 | 44.4 | 49.4 | 46.8 |
| OpenCon Sun & Li (2022) | 54.7 | 63.8 | 54.7 | 49.1 | 78.6 | 32.7 | - | - | - | - |
| PromptCAL Zhang et al. (2023) | 62.9 | 64.4 | 62.1 | 50.2 | 70.1 | 40.6 | 52.2 | 52.2 | 52.3 | 55.1 |
| DCCL Pu et al. (2023) | 63.5 | 60.8 | 64.9 | 43.1 | 55.7 | 36.2 | - | - | - | - |
| GPC Zhao et al. (2023) | 52.0 | 55.5 | 47.5 | 38.2 | 58.9 | 27.4 | 43.3 | 40.7 | 44.8 | 44.5 |
| SimGCD Wen et al. (2023) | 60.3 | 65.6 | 57.7 | 53.8 | 71.9 | 45.0 | 54.2 | 59.1 | 51.8 | 56.1 |
| μGCD Vaze et al. (2023) | 65.7 | 68.0 | 64.6 | 56.5 | 68.1 | 50.9 | 53.8 | 55.4 | 53.0 | 58.7 |
| InfoSieve Rastegar et al. (2023) | **69.4** | **77.9** | **65.2** | 55.7 | 74.8 | 46.4 | 56.3 | 63.7 | 52.5 | 60.5 |
| CiPR Hao et al. (2024) | 57.1 | 58.7 | 55.6 | 47.0 | 61.5 | 40.1 | - | - | - | - |
| SPTNet Wang et al. (2024b) | 65.8 | 68.8 | 65.1 | 59.0 | 79.2 | 49.3 | 59.3 | 61.8 | 58.1 | 61.4 |
| **DebGCD** | 66.3 | 71.8 | 63.5 | **65.3** | **81.6** | **57.4** | **61.7** | **63.9** | **60.6** | **64.4** |

Table 3: Comparison of state-of-the-art GCD methods on generic datasets. It includes CIFAR-10 Krizhevsky et al. (2009), CIFAR-100 Krizhevsky et al. (2009), ImageNet-100 Deng et al. (2009), and ImageNet-1K Deng et al. (2009) dataset.

| Method | CIFAR-10 | | | CIFAR-100 | | | ImageNet-100 | | | ImageNet-1K | | |
| --- | --- | --- | --- | --- | --- | --- | --- | --- | --- | --- | --- | --- |
| | All | Old | New | All | Old | New | All | Old | New | All | Old | New |
| k-means MacQueen (1967) | 83.6 | 85.7 | 82.5 | 52.0 | 52.2 | 50.8 | 72.7 | 75.5 | 71.3 | - | - | - |
| RankStats+ Han et al. (2021) | 46.8 | 19.2 | 60.5 | 58.2 | 77.6 | 19.3 | 37.1 | 61.6 | 24.8 | - | - | - |
| UNO+ Fini et al. (2021) | 68.6 | **98.3** | 53.8 | 69.5 | 80.6 | 47.2 | 70.3 | **95.0** | 57.9 | - | - | - |
| ORCA Cao et al. (2022) | 69.0 | 77.4 | 52.0 | 73.5 | **92.6** | 63.9 | 81.8 | 86.2 | 79.6 | - | - | - |
| GCD Vaze et al. (2022b) | 91.5 | 97.9 | 88.2 | 73.0 | 76.2 | 66.5 | 74.1 | 89.8 | 66.3 | 52.5 | 72.5 | 42.2 |
| XCon Fei et al. (2022) | 96.0 | 97.3 | 95.4 | 74.2 | 81.2 | 60.3 | 77.6 | 93.5 | 69.7 | - | - | - |
| OpenCon Sun & Li (2022) | - | - | - | - | - | - | 84.0 | 93.8 | 81.2 | - | - | - |
| PromptCAL Zhang et al. (2023) | **97.9** | 96.6 | 98.5 | 81.2 | 84.2 | 75.3 | 83.1 | 92.7 | 78.3 | - | - | - |
| DCCL Pu et al. (2023) | 96.3 | 96.5 | 96.9 | 75.3 | 76.8 | 70.2 | 80.5 | 90.5 | 76.2 | - | - | - |
| GPC Zhao et al. (2023) | 90.6 | 97.6 | 87.0 | 75.4 | 84.6 | 60.1 | 75.3 | 93.4 | 66.7 | - | - | - |
| SimGCD Wen et al. (2023) | 97.1 | 95.1 | 98.1 | 80.1 | 81.2 | 77.8 | 83.0 | 93.1 | 77.9 | 57.1 | 77.3 | 46.9 |
| InfoSieve Rastegar et al. (2023) | 94.8 | 97.7 | 93.4 | 78.3 | 82.2 | 70.5 | 80.5 | 93.8 | 73.8 | - | - | - |
| CiPR Hao et al. (2024) | 97.7 | 97.5 | 97.7 | 81.5 | 82.4 | 79.7 | 80.5 | 84.9 | 78.3 | - | - | - |
| SPTNet Wang et al. (2024b) | 97.3 | 95.0 | 98.6 | 81.3 | 84.3 | 75.6 | 85.4 | 93.2 | 81.4 | - | - | - |
| **DebGCD** | 97.2 | 94.8 | 98.4 | **83.0** | 84.6 | 79.9 | **85.9** | 94.3 | 81.6 | 65.0 | 82.0 | 56.5 |

dataset, while ranking second on CUB, where it is outperformed only by InfoSieve Rastegar et al. (2023), a hierarchical encoding method specifically designed for fine-grained GCD. In contrast, DebGCD aims for broader improvements across both generic and fine-grained datasets. These results reveal DebGCD's exceptional ability to uncover new categories, while also showcasing remarkable performance in recognizing known categories.

**Results on generic datasets.** In Tab. 3, we report results on three widely used generic datasets (CIFAR-10, CIFAR-100 and ImageNet-100) in GCD, as well as the challenging ImageNet-1K. Our method attains superior performance in terms of ACC across 'All' categories, establishing the new state-of-the-art, except CIFAR-10, on which the performance is nearly saturated (over 97% ACC) for our method and other most competitive methods. On the challenging ImageNet-1K, containing 1,000 classes with diverse images, DebGCD also establishes the new state-of-the-art, surpassing the previous best-performing method by 7.9%. These results validate the effectiveness and robustness of our method for generalized category discovery on generic datasets.

## 5.3 ANALYSIS

In this section, we provide ablations regarding the key components within our framework. Besides, we study the impact of the debiasing threshold $\tau$ and labelled data.

**Framework components.** Starting with the baseline method trained using $\mathcal{L}_{gcd}$ (Row (1)), we gradually incorporate our proposed techniques on the Stanford Cars dataset, as depicted in Tab. 4. An intuitive approach is to apply debiased learning to the original classifier as in Row (2). However, this still produces a biased supervision signal because it relies on the original GCD loss for that

Table 4: Ablations. The results regarding the different components in our framework on Stanford Cars Krause et al. (2013). ACC of 'All', 'Old' and 'New' categories are listed.

| | Debiased Learning | Auxiliary Classifier | Semantic Dist. Learning | Dist. Guidance | Stanford Cars | | |
|---|---|---|---|---|---|---|---|
| | | | | | All | Old | New |
| (1) | ✗ | ✗ | ✗ | ✗ | 53.8 | 71.9 | 45.0 |
| (2) | ✓ | ✗ | ✗ | ✗ | 51.3 | 72.8 | 40.9 |
| (3) | ✓ | ✓ | ✗ | ✗ | 58.5 | 78.7 | 48.8 |
| (4) | ✗ | ✗ | ✓ | ✗ | 56.5 | 73.3 | 48.3 |
| (5) | ✓ | ✓ | ✓ | ✗ | 60.7 | 78.1 | 52.3 |
| (6) | ✓ | ✓ | ✓ | ✓ | **65.3** | **81.6** | **57.4** |

classifier. It turns out that such a naïve approach may even hurt the performance. Rows (1) and (2) indicate that directly applying debiased learning to the GCD classifier can lead to a decrease in performance, particularly affecting novel categories. The introduction of an auxiliary classifier in Row (3) demonstrates significant performance enhancements. Similarly, our semantic distribution learning alone results in a 2.7% improvement across all categories in Row (4). Row (5) highlights that co-training the debiased classifier and semantic distribution detector further boosts performance. Notably, guiding the debiased learning with semantic distribution certainty and task consistency function yields a notable 4.6% performance increase in Row (6).

**Loss function.** In addition, we explore the impact of the data and the respective loss functions employed during the training of debiased classifier, denoted as $\mathcal{L}_{adl}^s$ and $\mathcal{L}_{adl}^u$, targeting the labelled and unlabelled datasets, respectively. The results are shown in Tab. 5. These experiments are undertaken on the FGVC-Aircraft Maji et al. (2013) using various subset combinations. Solely training with $\mathcal{L}_{adl}^s$ introduces bias towards known categories, leading to a notable performance decline. Conversely, exclusive training with $\mathcal{L}_{adl}^u$ fails to reach optimal performance levels, underscoring the essential role of knowledge derived from labelled data. These outcomes demonstrate the vital significance of both $\mathcal{L}_{adl}^s$ and $\mathcal{L}_{adl}^u$ in optimizing the debiased classifier.

Table 5: Experimental results on distillation data by using different loss functions.

| $\mathcal{L}_{adl}^s$ | $\mathcal{L}_{adl}^u$ | FGVC-Aircraft | | |
|---|---|---|---|---|
| | | All | Old | New |
| | | 54.2 | 59.1 | 51.8 |
| ✓ | | 53.1 | 60.5 | 49.4 |
| | ✓ | 57.9 | 60.1 | 56.9 |
| ✓ | ✓ | **61.7** | **63.9** | **60.6** |

**Debiasing threshold $\tau$.** Similar to self-training approaches Sohn et al. (2020); Zhang et al. (2021), the selection of the threshold for generating pseudo-labels also plays a crucial role in our approach. Consistent with the methods outlined in Wen et al. (2023) and Vaze et al. (2022b), we calibrate the threshold based on its performance on a separate validation set of the labelled data. Detailed results regarding different thresholds on the FGVC-Aircraft Wah et al.

Table 6: Experimental results regarding threshold $\tau$ on the unlabelled set and validation set of FGVC-Aircraft Maji et al. (2013) dataset.

| | Unlabelled Set | | | Validation Set | | |
|---|---|---|---|---|---|---|
| $\tau$ | All | Old | New | All | Old | New |
| 0.90 | 59.4 | **64.7** | 56.7 | 58.9 | 61.1 | 56.8 |
| 0.85 | **61.7** | 63.9 | **60.6** | **61.1** | **62.0** | **60.3** |
| 0.80 | 60.7 | 61.5 | 60.3 | 60.6 | 61.6 | 59.6 |

(2011) dataset, covering performance on both the unlabelled training dataset and the validation set, are presented. As shown in Tab. 6, the threshold is incrementally adjusted in intervals of 0.05. Notably, the performance trends for both datasets align, with optimal performance achieved when the threshold is set to 0.85.

# 6 CONCLUSION

This paper presents DebGCD, a distribution-guided debiased learning framework for GCD, comprising three primary components. Firstly, we introduce an auxiliary debiased learning mechanism by concurrently training a parallel classifier with the GCD classifier, thereby facilitating optimization in the GCD feature space. Secondly, a semantic distribution detector is introduced to explicitly identify semantic shifts and implicitly enhance performance. Lastly, we propose a semantic distribution certainty score that enables a curriculum-based learning approach, promoting effective learning for both seen and unseen classes. Despite its simplicity, DebGCD showcases superior performance, as evidenced by comprehensive evaluation on seven public benchmarks.

**Acknowledgements.** This work is supported by National Natural Science Foundation of China (Grant No. 62306251), Hong Kong Research Grant Council - Early Career Scheme (Grant No. 27208022), and HKU Seed Fund for Basic Research.

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
