# OpenReview forum: "DebGCD: Debiased Learning with Distribution Guidance for Generalized Category Discovery"
_ICLR.cc/2025/Conference — ICLR 2025 Poster_

### Official Review · Reviewer_Q4NM · 2024-11-01

**Soundness:** 3
**Presentation:** 3
**Contribution:** 2
**Rating:** 6
**Confidence:** 3

**Summary:**

The authors propose a novel framework called Debiased Learning with Distribution Guidance (D2G) for the GCD task, which introduces a debiased learning paradigm to optimize the clustering feature space and learns a semantic distribution detector to enhance the learning effect of GCD. Besides, D2G propose a curriculum learning mechanism that steers the debiased learning process to effectively mitigate the negative impact of uncertain samples. The authors evaluate the method on the public GCD benchmarks to demonstrate the effectiveness.

**Strengths:**

D2G considers both label bias and semantic shift to address the challenging GCD task. It’s a novel idea to mark the first exploration of these aspects.
D2G effectively incorporates all components into a unified framework and can be trained in a single stage without any additional computation burden.
The authors conduct extensive experimentation on public GCD benchmarks to demonstrate its effectiveness.

**Weaknesses:**

1. The reason for using OOD techniques to solve GCD task is not clear because the objectives of these two tasks are different. The motivation for using MLP projection network to solve this problem needs further explanation.
2. From the experimental results, the performance improvement of the method is not significant, especially on the CUB dataset. Besides, there are few comparison methods on the ImageNet-1K dataset, which can lead to unreliable comparison results.
3. Some hyperparameters lack ablation experiments to verify that the experimental method is optimal, including the number of layers in MLPs, the loss weights, and so on.

**Questions:**

1. I wonder whether the GCD method is sensitive to certain categories, resulting in limited performance improvement on some datasets. Perhaps the authors can design some experiments to test it.
2. Is it better to use vision-language pre-trained large models such as CLIP to solve label bias problems, as CLIP contains a lot of pre-trained knowledge for new categories.

---

> ### Author Response · Authors · 2024-11-25
> **Response to Reviewer Q4NM(1/2)**
>
> ### **Q1. Reason for using OOD techniques to solve GCD**
> Thanks for your insightful comments. The motivation for utilizing OOD techniques stems from the inherent semantic shifts present in the GCD task, which involves both known and unknown classes within unlabelled data. Although the objectives of the GCD task and OOD techniques differ, OOD can supply valuable semantic distribution information that can guide the training of GCD on unlabelled data. Additionally, in other open-world tasks such as open-set semi-supervised learning [1] and universal domain adaptation [2], OOD techniques have been shown to offer significant advantages.
>
> ### **Q2. Motivation for using MLP**
> Thanks for your insightful comments. Please refer to Q1 in general response.
>
> ### **Q3. Performance improvement**
> Thanks for the comments. On the fine-grained SSB benchmark, our method achieves the highest *ACC*, clearly surpassing the previous SOTA (64.4 vs. 61.4, as shown in Tab.2). In terms of generic datasets, our method outperforms others on three out of four datasets regarding *ACC* on 'All' categories, being only 0.7% lower than the best performance on CIFAR10, where the results are nearly saturated.
>
> ### **Q4. Few comparison methods on ImageNet-1K**
> Thanks for your insightful comments. Indeed, very few methods have reported results on ImageNet-1K due to the high computational costs involved. We have carefully gathered all publicly available results for ImageNet-1K and found that only SimGCD has reported results, which are inferior to ours.
>
>
> ### **Q5. Ablation on the number of layers in MLP and loss weight**
> Thanks for your insightful comments. Please refer to Q1 and Q2 in general response.
>
> ### **Q6. Limited performance improvement on some datasets**
> Thanks for your insightful comments. To address this concern, we provide quantitative analysis on the  improvements brought by our method. Particularly, we examine the baseline model’s prediction by categorizing the errors into four types based on the relationship between the predicted and ground-truth classes: "True Old", "False New", "False Old", and "True New".  For example, "True New" refers to wrongly predicting a 'New' class sample to another 'New' class, while "False Old" indicates predicting a 'New' class sample as some 'Old' class.
>
> From this perspective, our debiased learning method primarily aims to mitigate the label bias between 'Old' and 'New' classes, thereby reducing the likelihood of 'New' class samples being predicted as 'Old'. Consequently, this reduction in bias leads to a decrease in "False Old" predictions and indirectly improves other types of error.
> In Table A10, we present the ratios of the four prediction error types for SimGCD with the DINO backbone across three datasets in the SSB benchmark.
> The results reveal that error distributions vary significantly across datasets, influenced by the dataset's characteristics, the classification of known and unknown categories, the baseline design, and the pretrained backbone network used. Notably, the Stanford Cars dataset exhibits the highest number of "False Old" samples, explaining why our method demonstrates the most substantial performance improvement on this dataset. In contrast, the CUB dataset shows the fewest "False Old" samples, indicating *relatively limited potential* for performance enhancement. We have incorporated this analysis into our revised version.
>
>
> **Table A10. Error analysis of SimGCD on different datasets in the SSB benchmark.**
> |CUB|Pred&nbsp;Old|Pred&nbsp;New|SCars|Pred&nbsp;Old|Pred&nbsp;New|Aircraft|Pred&nbsp;Old|Pred&nbsp;New|
> --------------|-------------|------------|------------|------------|------------|------------|------------|------------|
> | **GT&nbsp;Old**|3.2%|31.1%|**GT&nbsp;Old**|9.9%|18.1%|**GT&nbsp;Old**|13.7% |27.4%|
> | **GT&nbsp;New**|**8.0%**|35.0%|**GT&nbsp;New**| **16.5%**|38.8%|**GT&nbsp;New**|**10.4%** |37.9%|
>
>
>
> *[1] Yu, Qing, et al. "Multi-task curriculum framework for open-set semi-supervised learning." ECCV, 2020.*
>
> *[2] Saito, Kuniaki, and Kate Saenko. "Ovanet: One-vs-all network for universal domain adaptation." ICCV 2021.*

---

> ### Author Response · Authors · 2024-11-25
> **Response to Reviewer Q4NM(2/2)**
>
> ### **Q7. Using CLIP in GCD**
> Thanks for your insightful comments.
> **_Firstly_**, the label bias problem we aim to address arises from the design of previous GCD methods: unlike known categories, unknown categories only receive soft supervision (see Fig.1). In these methods, label bias remains an issue regardless of the pre-trained backbone network used.
> **_Secondly_**, there have indeed been attempts to leverage CLIP for tackling GCD, such as CLIP-GCD [1], and GET [2]. CLIP provides strong representation capabilities and demonstrates good zero-shot transfer performance. However, it requires expensive large-scale pretraining and poses a risk of data contamination, complicating the distinction between seen and unseen classes. Additionally, it struggles with instances that fall outside the text vocabulary, particularly with the emergence of previously unseen classes.
> **_Thirdly_**, despite these concerns, we can still try to apply CLIP to both our method and the baseline, and we observe consistent performance improvements with the DINO backbone. As shown in Table A11, our D2G achieves an average improvement of 8.2% in *ACC* across 'All' categories on the SSB benchmark, achieving a highest average *ACC*.
>
> **Table A11. GCD performance using CLIP.**
> | |CUB | SCars | Aircraft|Average
> |--------------|-------------|--------------|--------------|--------------
> |**Method**|**All/Old/New** | **All/Old/New**| **All/Old/New** |**All**
> |CLIP-GCD[1]|62.8/77.1/55.7|70.6/88.2/62.2|50.0/56.6/46.5|61.1
> |GET[2]|77.0/78.1/**76.4**|78.5/86.8/74.5|58.9/59.6/58.5|71.5
> |SimGCD-CLIP|69.8/75.5/67.0|71.8/81.7/67.0|56.3/61.1/53.9|66.0|
> |D2G-CLIP|**77.3**/**82.0**/74.9|**80.3**/**91.9**/**74.8**|**64.9**/**68.5**/**63.1**|**74.2**|
>
>
>
> *[1] Ouldnoughi, Rabah, Chia-Wen Kuo, and Zsolt Kira. "Clip-gcd: Simple language guided generalized category discovery." arxiv, 2023.*
>
> *[2] Wang, Enguang, et al. "GET: Unlocking the Multi-modal Potential of CLIP for Generalized Category Discovery." arxiv, 2024.*

---

> ### Author Response · Authors · 2024-11-29
> **A Kind Reminder for Reading the Response**
>
> Dear Reviewer Q4NM,
>
> We greatly appreciate your valuable time and effort in reviewing our paper. We understand that this may be a busy period for you. As the discussion phase draws to a close, we kindly request your feedback on our responses. If you have any additional comments or questions regarding our paper, we would be more than happy to discuss them with you in detail.
>
> We look forward to your reply.
>
> Best regards,
>
> Authors

---

> > ### Comment · Reviewer_Q4NM · 2024-12-02
> > **I have raised my rating**
> >
> > The authors have well addressed the issues I raised. So, I have improved my rating.

---

> > > ### Author Response · Authors · 2024-12-02
> > > **Reply to the feedback from Reviewer Q4NM**
> > >
> > > We are glad that our responses have effectively addressed your concerns. Thank you for your insightful comments and valuable contributions, which are instrumental in enhancing our paper.

---

### Official Review · Reviewer_59SW · 2024-11-02

**Soundness:** 2
**Presentation:** 3
**Contribution:** 3
**Rating:** 6
**Confidence:** 4

**Summary:**

This paper presents the D2G (Debiased Learning with Distribution Guidance) framework for addressing the Generalized Category Discovery (GCD) problem. GCD is challenging due to label biases and semantic shifts between known and unknown categories. The D2G framework introduces a debiased learning paradigm, a semantic distribution detector, and a curriculum learning approach based on distribution certainty to address these issues. Extensive experiments demonstrate D2G’s superiority over existing GCD methods on various benchmarks.

**Strengths:**

1. The paper is well-organized, with clear explanations of technical details.
2. The introduction of a debiased learning framework specific to GCD with a multi-feature distribution approach is innovative.
3. The technical contributions are well-structured and effectively evaluated. The integration of auxiliary debiased learning, semantic detection, and curriculum learning reinforces the model's performance.

**Weaknesses:**

1. The variation in results from the GCD benchmarks can be very large, so it is important to report all results as well as the error bars from the three independent runs, as SimGCD does in its Supplementary Information.
2. While the authors claim that D2G does not introduce additional computational burdens during inference, a more detailed analysis of the training time and computational costs associated with the auxiliary components would be valuable.

**Questions:**

1. What strategies do authors envision to reduce potential overfitting during assisted debiasing learning, especially when utilizing limited unlabeled data on fine-grained datasets?
2. Based on Table 4, it can be concluded that the effect of label debiasing is not very good. Have the authors considered not using the label debiasing strategy? For example, what are the results for "w/o debiased learning, w/ auxiliary classifier, w/o semantic dist. learning, w/o dist. guidance"?

---

> ### Author Response · Authors · 2024-11-25
> **Response to Reviewer 59SW**
>
> ### **Q1. Multi-run results**
> Thanks for your insightful comments. We have included multi-run results for CUB, Stanford Cars, FGVC-Aircraft, CIFAR-10, CIFAR-100, ImageNet-100, ImageNet-1K, and Herbarium19, as shown in Table A8. Despite achieving significantly higher performance, we observe that the variance is even smaller than that of SimGCD (refer to Tab.6 in the Appendix B.1 of the SimGCD paper). We have added this table to the Appendix in the revised version.
>
> **Table A8. Multi-run results of D2G.**
> | dataset|All|Old|New|
> |--------------|-------------|--------------|--------------
> |CUB|66.4±0.4|72.9±0.6|63.2±0.4|
> |Scars|65.2±0.7|81.7±1.2|57.3±0.6|
> |Aircraft|61.7±0.5|65.9±1.2|59.5±1.1|
> |CIFAR-10|97.3±0.1|95.0±0.2|98.4±0.1
> |CIFAR-100|83.1±0.7|84.7±0.7|80.0±0.9
> |ImageNet-100|86.1±0.6|94.5±0.5|81.8±0.6
> |ImageNet-1K|64.9±0.3|82.1±0.2|56.4±0.4
> |Oxford-Pet|93.2±0.2|86.3±0.1|96.8±0.3
> |Herbarium19|44.9±0.3|59.3±0.3|37.1±0.5
>
> ### **Q2. Training time and computational costs**
> Thanks for your insightful comments. We provide the following information below. During inference, all proposed components will be removed, resulting in the computational cost of D2G being equivalent to that of SimGCD. During training, since the main computational cost arises from the backbone network (ViT-B/16), the difference in training time and computational cost between SimGCD and our method is negligible when the same blocks are tuned. In the table below, we present the FLOPs required for a single forward step and the elapsed time for a single training iteration on the CUB dataset, utilizing a batch size of 128 with a NVIDIA V100 GPU. The performance comparison when tuning the same blocks can be found in Q3 of Reviewer YCCi.
>
> **Table A9. FLOPs and time costs during training.**
> | Method|FLOPs(G)|Time Cost Per Iteration(s)|
> |--------------|-------------|--------------
> |SimGCD(tune one block)|2159.3|1.081|
> |SimGCD(tune two blocks)|2159.3|1.253|
> |D2G(tune one block)|2161.2|1.082|
> |D2G(tune two blocks)|2161.2|1.256|
>
>
> ### **Q3. Strategies to reduce potential overfitting**
> Thanks for your insightful comments. We have incorporated several strategies in the baseline, such as entropy regularization and weight decay, which are effective in reducing overfitting. Therefore, we did not introduce any additional strategies for our debiased learning approach. As indicated by the performance on the held-out validation set  (see Fig.5 in the supplementary materials), the model did not exhibit overfitting to the training set.
>
>
> ### **Q4. The effect of label debiasing**
> Thanks for your insightful comments. We would like to clarify that while the label debiasing strategy is effective, the auxiliary classifier serves as our vehicle for implementing debiased learning (refer to Table 4 and lines 482–484 in the paper). As shown in Tab.4, applying the debiased loss to the original classifier leads to a decline in performance, as indicated in row 2 compared to row 1. This decline is primarily due to the reliance on the original GCD loss for that classifier, which still results in a biased supervision signal. Comparing row 3 with row 1, we observe a notable improvement of 4.7% in 'All' *ACC* over the baseline, achieved through our debiased learning with the auxiliary classifier. This highlights the necessity of our auxiliary classifier, which is designed to be debiased in order to facilitate effective debiased learning and optimize the shared GCD feature space. Furthermore, if we do not utilize the label debiasing strategy, the auxiliary classifier—being a core component of this strategy—should be excluded entirely. We have included this clarification in the revised version.

---

> > ### Comment · Reviewer_59SW · 2024-11-28
> >
> > I appreciate the authors' response, and I will maintain the initial score.

---

> > > ### Author Response · Authors · 2024-11-29
> > > **Reply to the feedback from Reviewer 59SW**
> > >
> > > Thank you for your recognition and kind words regarding our work. Your appreciation inspires us a lot.

---

### Official Review · Reviewer_M5TQ · 2024-11-03

**Soundness:** 3
**Presentation:** 2
**Contribution:** 2
**Rating:** 6
**Confidence:** 4

**Summary:**

This paper propose D2G a novel framework that addresses the challenging GCD task. Several new paradigms and mechanisms like debias learning in this framework enhance the model’s performance. Combined with these, the method proposed demonstrates its effectiveness and achieves superior performance on broad benchmark.

**Strengths:**

1. The paper writing is clear and easy to understand.
2. The topic of the article is of significant theoretical and practical importance, addressing a gap in the existing literature.
3. The paper clearly outlines the shortcomings of previous studies and results section is logically organized.
4. The proposed framework for DCG is novelty. Various incremental mechanisms make sense to me.

**Weaknesses:**

1. There is a error in fig.1 (d), the brown dish line is invisible.
2. The hyperparameters in Eq. 14 are empirical values or obtained through experiments? If latter, I believe the authors should include some ablation studies for clarification.
3. In Tab. 2, the performance of D2F is suboptimal compared to InfoSieve, but it lacks a specific analysis. Could the authors provide further details on this?
4. The impact of the various method proposed, such as Debias learning and Auxiliary Classifier, should be evaluated through ablation studies on a broader dataset. The paper currently reports results only on the Stanford dataset. Do authors validate only on this dataset?

**Questions:**

please refer to weakness.

---

> ### Author Response · Authors · 2024-11-25
> **Response to Reviewer M5TQ**
>
> ### **Q1. Brown dashed line in Fig.1**
> Thank you for pointing this out. However, we have observed that the document displays normally on our end, across various PDF readers and browsers. We would appreciate it if the reviewer could provide more details about the issue encountered, as we would be happy to investigate and work on a resolution.
>
> ### **Q2. Hyperparameters in Eq.14**
> Thanks for your insightful comments. Please refer to Q2 in general response.
>
>
> ### **Q3. Comparison with InfoSeive**
> Thanks for your insightful comments. Infosieve is a hierarchical encoding method specifically designed for fine-grained GCD, which may work well for certain datasets. In contrast, our method does not incorporate specific designs tailored for fine-grained datasets; instead, it aims for broader improvements across both generic and fine-grained datasets. Notably, our method significantly outperforms Infosieve on the SSB benchmark (64.4 vs. 60.5, see Tab.2), showing only a slight performance gap (3.1% lower on 'All' categories) on one of the three datasets. On Stanford Cars and FGVC-Aircraft, our method demonstrates considerable advantages, achieving improvements of 9.6% and 5.4% on 'All' categories, respectively. Additionally, on all generic datasets (see Tab.3), our method consistently surpasses Infosieve, with improvements of 2.4%, 4.7%, and 5.4% on CIFAR10, CIFAR100, and ImageNet-100, respectively, on the 'All' categories.
>
>
> ### **Q4. Ablation studies on more datasets**
> Thanks for your insightful comments. We have included additional ablation results for the other two datasets in the SSB benchmark (CUB and FGVC-Aircraft), as well as the generic dataset ImageNet-100, in Table A7. In this table, the letters a, b, c, and d represent Debiased Learning, Auxiliary Classifier, Semantic Distribution Learning, and Distribution Guidance, respectively. Our results indicate that directly applying debiased learning to the original GCD classifier may lead to a drop in performance (see comparisons between (1) and (2)). In contrast, when using an auxiliary classifier, we observe an improvement in performance (comparing (1) and (3)). Furthermore, the joint training of the debiased classifier and the OOD detector yields additional enhancements (comparing (3) and (5)). Finally, the introduction of distribution guidance leads to further performance improvements. These findings are consistent with those obtained on the Stanford Cars dataset, as shown in Tab.4 of our paper. We have incorporated these results into the revised version, as suggested.
>
> **Table A7. Additional ablation studies on CUB, Aircraft and ImageNet-100.**
> || | | | |CUB | Aircraft| IN-100|
> |--------------|--------------|-------------|--------------|--------------|-------------|--------------|--------------
> ||**a**|**b**|**c**|**d**|**All/Old/New** | **All/Old/New** | **All/Old/New** |
> (1)|||||60.3/65.6/57.7|54.2/59.1/51.8|83.0/93.1/77.9|
> (2)|✔||||58.6/72.3/51.7|53.7/62.9/49.1|82.8/94.1/77.2|
> (3)|✔|✔|||63.8/69.3/61.1|57.7/59.8/56.5|84.7/94.0/80.0|
> (4)|||✔||61.3/69.4/57.3|56.6/64.8/52.5|83.5/92.4/78.9|
> (5)|✔|✔|✔||64.9/70.9/61.9|59.4/**64.4**/56.9|85.0/93.8/80.3|
> (6)|✔|✔|✔|✔|**66.3/71.8/63.5**|**61.7**/63.9/**60.6**|**85.9/94.3/81.6**|

---

> > ### Comment · Reviewer_M5TQ · 2024-11-28
> >
> > Thank you for the detailed response from the author. I will maintain my score.

---

> > > ### Author Response · Authors · 2024-11-29
> > > **Reply to the feedback from Reviewer M5TQ**
> > >
> > > Thank you for your recognition and kind words regarding our work. Your appreciation truly inspires us.

---

### Official Review · Reviewer_YCCi · 2024-11-03

**Soundness:** 3
**Presentation:** 3
**Contribution:** 3
**Rating:** 6
**Confidence:** 5

**Summary:**

The paper claims that existing GCD methods suffer from label bias, fail to account for differences in uncertainty, and do not address semantic distribution shifts. To address these issues, the author proposes D2G framework, which comprises Semantic Distribution Detection and Auxiliary Debiased Learning. The Semantic Distribution Detection module treats each labeled category as a separate binary classification, using the prediction confidence score obtained to filter and scale the debiased loss. The additional loss introduced by these components can be directly integrated with SimGCD and these modules can be entirely discarded during inference.

**Strengths:**

1. The motivation behind addressing label bias is sound to me. Previous methods apply soft supervision to unlabeled data, which results in weaker supervision for unknown classes. The proposed method aligns well with this motivation.
2. The approach achieves performance improvements demonstrating its effectiveness.
3. The framework is efficient in inference.
4. The writing is clear and easy to follow.

**Weaknesses:**

1. The distribution detector functions as multiple independent binary classification, so there is no competition between categories. It serves two purposes: first, it uses negative class confidence scores to filter out likely unknown classes in the final $L^u_{adl}$; second, it imposes stronger supervision on samples with higher uncertainty. For the first purpose, is there a significant difference in effectiveness compared to using self-entropy to filter unknown samples? Self-entropy would seem a more natural and straightforward metric, yet the author does not analyze the benefits of this one-vs-all design. For the second, Equation 10 imposes stronger pseudo one-hot supervision on samples deemed uncertain by the distribution detector. For example, if an unknown class is close to a known class, the loss will be reduced by $d_i$. The ablation study indicates that this yields significant performance gains, but lacks detailed analysis and discussion.
2. The D2G framework finetunes more parameters than SimGCD, which only trains the last block. Since D2G builds on SimGCD, it would be more meaningful to compare performance under the same training setup. The authors did not provide this.
3. There is a performance drop compared to the baseline on Herbarium19.
4. Since all introduced modules can be discarded during inference, I think the key to performance improvement likely lies in the enhancement of the discriminability of DINO CLS token. However, the authors provide minimal discussion on this aspect.
5. The description of the ablation studies is not sufficiently clear, and there is a lack of discussion between experiments. Specifically, regarding debiased learning, all debiased losses could theoretically be applied directly to the original classifier in SimGCD. It is unclear why the addition of a second classifier is necessary for effective performance. Additionally, I would like to know the impact on performance of removing the MLP prior to the OVA module.

**Questions:**

See weakness.

---

> ### Author Response · Authors · 2024-11-25
> **Response to Reviewer YCCi(1/2)**
>
> ### **Q1. OVA vs self-entropy**
> Thanks for your insightful comments.
> **_Firstly_**, we agree that self-entropy may seem like a more intuitive approach than the OVA design for OOD detection. In common practice, the maximum score or logit on categories from a closed-set classifier can serve as a good indicator of OOD. However, the situation is different for the GCD classifier. There is an entropy regularization term in the loss function (see $H(\overline{p})$ in Eq.2), which aims to prevent trivial predictions. The SimGCD paper indicates that this regularization term helps mitigate prediction bias between seen and novel classes. Nevertheless, we find that it also results in the classifier's predictions on known categories being less confident than those of a closed-set classifier, thereby degrading the OOD detection performance of the GCD classifier.
> **_Secondly_**, our framework does not aim to strictly partition the in-distribution (ID) and OOD samples, as such strict separation could introduce errors to training. Instead, our goal is to utilize the distribution certainty of each sample. A significant drawback of self-entropy-based OOD methods is the necessity to manually establish a threshold for rejecting "unknown" samples, which relies on validation or a pre-defined ratio of "unknown" samples. This approach is impractical and complicates the definition of unified certainty scores for all samples. In contrast, OVA-based methods eliminate the need for threshold searching, allowing us to simply use a threshold of 0.5, which facilitates the definition of a simple certainty score as presented in Eq. 10.
> **_Thirdly_**, to further validate the above hypothesis, we evaluated the OOD performance of the GCD classifier by using the maximum score on known categories as the ID score. The AUROC results are shown below. Compared to the results in Tab.13, we observe that this approach is less effective than the OVA classifier, as shown in Table A5. Moreover, when we train the OVA classifier, debiased classifier, and GCD classifier simultaneously, the three tasks can mutually benefit both OOD and GCD performance, as illustrated in Tab.4(row 1 & row 4) and Tab.13(row 1 & row 2) in the paper. We have included this discussion in the revised version.
>
> **Table A5. The OOD performance of different methods.**
> | |CIFAR10|CIFAR100|IN-100| CUB | SCars | Aircraft|
> |--------------------------|----------------|--------------|--------------|--------------|--------------|--------------
> |self-entropy|70.3|85.2|91.7|71.8|73.2|73.1|
> |OVA|66.1|90.8|96.5|77.5|78.6|76.2|
> |OVA+ours|97.5|94.8|99.5|86.8|89.6|86.3|
>
>
> ### **Q2. Supervision on uncertain samples**
> Thanks for the comments. We would like to clarify that Eq. 10 indeed imposes weaker supervision for samples identified as uncertain by the OOD detector. For these uncertain samples, their OOD score $s_i$ will be close to 0.5, resulting in their distribution certainty score $d_i$ approaching 0. Consequently, Eq. 10 imposes a weaker pseudo one-hot supervision on these samples.
>
> ### **Q3. Performance comparison with the same tuned blocks**
> Thanks for your insightful comments. We would like to present the performance of training the last one or two blocks. As demonstrated in Table A6, this does not result in significant improvements for SimGCD. In contrast to SimGCD, our framework incorporates additional tasks, including OOD detection and debiased learning. We empirically observed that increasing the number of trainable parameters can improve performance on specific datasets, particularly those that are fine-grained. Similar strategies have been employed in previous methods, such as Infosieve[1]. We have included this comparison in the revised version.
>
> **Table A6. Performance comparison of SimGCD and D2G with different tuned blocks.**
> | || CUB | SCars | Aircraft|IN-100|CIFAR100
> |--------------|--------------|--------------|--------------|--------------|--------------|--------------
> |**Method**|**Setting**| **All/Old/New** | **All/Old/New**| **All/Old/New** |**All/Old/New**|**All/Old/New**
> |SimGCD|tune&nbsp;one&nbsp;block|60.3/65.6/57.7|53.8/71.9/45.0|54.2/59.1/51.8|83.0/93.1/77.9|80.1/81.2/77.8
> |SimGCD|tune&nbsp;two&nbsp;blocks|60.8/65.8/58.4|53.6/67.6/49.8|52.8/56.8/50.8|83.2/92.9/78.3|79.4/80.1/77.3
> |D2G|tune&nbsp;one&nbsp;block|65.1/70.9/62.2|63.0/80.2/54.7|60.4/**65.0**/58.1|85.7/94.0/81.5|82.4/83.6/79.5
> |D2G|tune&nbsp;two&nbsp;blocks|**66.3/71.8/63.5**|**65.3/81.6/57.4**|**61.7**/63.9/**60.6**|**85.9/94.3/81.6**|**83.0/84.6/79.9**

---

> ### Author Response · Authors · 2024-11-25
> **Response to Reviewer YCCi(2/2)**
>
> ### **Q4. Performance on Herbarium19**
> Thanks for the comments. In fact, there is a noticeable performance improvement (see Tab.7) over SimGCD baseline. Our method achieves an _ACC_ of (44.7, 59.4, 36.8), while SimGCD records an _ACC_ of (44.0, 58.0, 36.4) for the 'All', 'Old', and 'New' categories, respectively. It is important to highlight that the Herbarium19 dataset poses unique challenges due to its long-tailed characteristics, which complicates performance for both SimGCD and our method.
>
>
> ### **Q5. Key to performance improvement**
> Thanks for your insightful comments. We agree that the *CLS* token plays a crucial role in enhancing our performance. This improvement is achieved through the optimization of the entire embedding space and classifier via our debiased learning approach. In Fig.7, we visualize the cross-attention between the *CLS* token and patch embeddings, revealing that the maps generated by our model predominantly focus on the object of interest while effectively ignoring spurious factors and background clutter. The attention maps demonstrate that the *CLS* tokens in our method exhibit stronger discriminative power compared to the patch tokens, thereby validating that *CLS* tokens are more effective for distinguishing between both seen and unseen classes. We have incorporated this explanation in the revised version as suggested.
>
> ### **Q6. Debiased loss on the original GCD classifier**
> Thank you for raising this question. Indeed, we considered this solution at the very beginning of our project. However, we found that applying the debiased loss to the original classifier still results in a biased supervision signal, primarily due to the reliance on the original GCD loss for that classifier. In fact, incorporating the debiased loss into the original classifier leads to a decline in performance (see row 1 & row 2 in Tab.4). This highlights the need for a second classifier, which is debiased by design, in order to facilitate debiased learning and optimize the shared GCD feature space. We have included this explanation in the revised version as suggested.
>
> ### **Q7. Impact of MLP prior to OVA module**
> Thanks for your insightful comments. Please refer to Q1 in general response.
>
> *[1] Rastegar, Sarah, Hazel Doughty, and Cees Snoek. "Learn to categorize or categorize to learn? self-coding for generalized category discovery." NeurIPS 2024.*

---

> ### Author Response · Authors · 2024-11-29
> **A Kind Reminder for Reading the Response**
>
> Dear Reviewer YCCi,
>
>
> We greatly appreciate your valuable time and effort in reviewing our paper. We understand that this may be a busy period for you. As the discussion phase draws to a close, we kindly request your feedback on our responses. If you have any additional comments or questions regarding our paper, we would be more than happy to discuss them with you in detail.
>
>
> We look forward to your reply.
>
>
> Best regards,
>
> Authors

---

> > ### Comment · Reviewer_YCCi · 2024-11-29
> > **Official Comment by Reviewer YCCi**
> >
> > Thank you for the detailed response. Upon revisiting the revised version of the paper, I realize that my previous understanding of certain details was incorrect. The authors' response addresses all my concerns thoroughly. Therefore, I decide to raise my score.

---

> > > ### Author Response · Authors · 2024-11-30
> > > **Reply to the feedback from Reviewer YCCi**
> > >
> > > Thank you for your insightful feedback. Your appreciation truly means a great deal to us.

---

### Author Response · Authors · 2024-11-25
**General response(1/2)**

We thank all the reviewers for their insightful comments and positive feedback. Reviewer YCCi marked that **"the motivation behind addressing label bias is sound to me,"** and noted that **"the proposed method aligns well with this motivation"** and that **"the framework is efficient in inference."** Reviewer M5TQ commented that our work is **"addressing a gap in the existing literature"** and highlighted that **"the proposed framework for GCD is novel."** Additionally, Reviewer 59SW stated that **"the introduction of a debiased learning framework specific to GCD with a multi-feature distribution approach is innovative"** and affirmed that **"the technical contributions are well-structured and effectively evaluated."** Reviewer Q4NM also noted that **"it's a novel idea."** Furthermore, the reviewers agreed that our paper is **"clear and easy to follow"** (Reviewer YCCi), **"clear and easy to understand"** (Reviewer M5TQ), and **"well-organized, with clear explanations of technical details"** (Reviewer 59SW).

We have carefully addressed all concerns raised by the reviewers. First, we provide a **general response** to the shared concerns and critical points. We then address the individual concerns of each reviewer following their comments. We will also further strengthen the final manuscript based on the reviewers' concluding feedback.

**Code:** A well-documented code together with all trained models will be made public.

---

> ### Author Response · Authors · 2024-11-27
> **Summary of revisions**
>
> We have revised the paper and would like to invite the reviewers to take a look. Following the reviewers' suggestions, we have made the following major updates:
>
> **Section 4.1**: Added a detailed discussion about self-entropy and OVA for OOD scenarios, along with a clearer motivation for using the MLP.
>
> **Section 5.2**: Provided additional discussion with InfoSieve.
>
> **Section 5.3**: Expanded the discussion on debiased learning for the GCD classifier.
>
> **Appendix I**: Included a more comprehensive discussion about the CLS token.
>
> **Appendix J**: Added ablation studies conducted on additional datasets.
>
> **Appendix K**: Provided hyperparameter analysis regarding the number of MLP layers, loss weights, and the number of tuned blocks.
>
> **Appendix L**: Included results from multi-run experiments.
>
> **Appendix M**: Added an analysis of prediction errors.
>
> Please let us know if there are any further concerns.

---

### Author Response · Authors · 2024-11-25
**General response(2/2)**

### **Q1. Motivation and  ablation about the MLP before OVA (Reviewers YCCi and Q4NM)**
We appreciate the reviewers' suggestions and have incorporated the requested experiments. In the context of OOD, our objective is not to differentiate between multiple distinct unknown categories, as in GCD; rather, we aim to distinguish all unknown samples from the known classes, effectively framing this as a binary classification problem. This requirement led us to introduce a different embedding space that is more suitable for this task, achieved by simply adding an MLP projection head.
To validate the impact of the number of layers in the MLP, we conducted an ablation study on the SSB benchmark regarding GCD and OOD performance, as shown in the Table A1 and Table A2. We observe that the average GCD performance across all categories of D2G gradually improves as the number of MLP layers increases from 0 to 5. A similar trend is evident in the OOD performance. However, extending the MLP to 7 layers results in little to no improvement in performance. In our implementation, we adopted a 5-layer MLP in our framework.


**Table A1. The GCD performance using different number of MLP layers.**
| | CUB | SCars | Aircraft|Average
|--------------|--------------|--------------|--------------|--------------
**MLP layer**| **All/Old/New** | **All/Old/New**| **All/Old/New** |**All**
0|63.6/75.2/57.8|62.3/76.2/54.1|59.6/62.2/58.3|61.8
1|64.9/71.6/61.6|63.9/80.2/56.0|60.7/63.7/59.2|63.1
3|66.0/**73.5**/62.3|64.7/**82.2**/56.2|61.1/64.2/59.5|63.9
5|**66.3**/71.8/**63.5**|**65.3**/81.6/**57.4**|61.7/63.9/**60.6**|**64.4**
7|65.8/72.0/62.7|64.8/80.5/57.3|**61.9**/**65.2**/60.3|64.1

**Table A2. The OOD performance using different number of MLP layers.**
| | CUB | SCars | Aircraft|Average
|--------------|--------------|--------------|--------------|--------------
**MLP layer**| **AUROC** | **AUROC**| **AUROC** |**AUROC**
0|80.1|81.3|79.0|80.1
1|84.2|85.1|83.5|84.2
3|86.9|87.4|85.9|86.7
5|86.8|**89.6**|86.3|87.6
7|**87.2**|89.4|**87.1**|**87.9**


### **Q2. Further ablation about the loss weights (Reviewers M5TQ and Q4NM)**
We appreciate the suggestions from the reviewers and have included the requested experiments. Indeed, we do not extensively tune the hyperparameters but intuitively set them based on existing literature and our hypothesis. Our rationale for selecting values for the loss weights is as follows:
For $\lambda_{sdl}$, we take inspiration from the previous literature using OVA classifier[1]. In the paper, the model is fine-tuned with a learning rate of $10^{-3}$ , while the learning rate in the SimGCD baseline is 0.1 (which is 100 times larger than $10^{-3}$). To achieve a similar learning effect, as validated in OVA, we scale our $\lambda_{sdl}$ value from 1 down to (1/100). Therefore, we set $\lambda_{sdl} = 0.01$ by default.
For $\lambda_{adl}$, the weight of the debiased classifier, we expect it to play an important role similar to that of the original GCD classifier (where the loss weight is set to 1). Thus, we have defaulted this value to 1.0.
After determining the default values, we conducted experiments on the SSB benchmark regarding the two loss weights by exploring values around the defaults. For $\lambda_{sdl}$, the range was (0.005, 0.01, 0.02). As for $\lambda_{adl}$, the range was (0.5, 1.0, 2.0). The impact of $\lambda_{sdl}$ is detailed below in Table A3, with $\lambda_{adl}$ set to 1.0. The impact of $\lambda_{adl}$ is illustrated below in Table A4, with $\lambda_{sdl}$ set to 0.01. The results are in line with our hypothesis, indicating that our selected hyperparameters are indeed reasonable.

**Table A3. The ablation results about $\lambda_{sdl}$ on the SSB benchmark.**
||CUB | SCars | Aircraft|Average
|--------------|-------------|--------------|--------------|--------------
|$\lambda_{sdl}$| **All/Old/New** | **All/Old/New**| **All/Old/New** |**All**
|0.02|65.5/**73.2**/61.6|64.3/79.2/57.1|60.6/63.5/59.1|63.5
|0.01|**66.3**/71.8/**63.5**|**65.3/81.6/57.4**|61.7/63.9/**60.6**|**64.4**
|0.005|65.8/72.4/62.5|64.9/81.2/57.0|**62.1/65.4**/60.3|64.3


**Table A4. The ablation results about $\lambda_{sdl}$ on the SSB benchmark.**
||CUB | SCars | Aircraft|Average
|--------------|-------------|--------------|--------------|--------------
|$\lambda_{adl}$| **All/Old/New** | **All/Old/New**| **All/Old/New** |**All**
|0.5|64.3/**72.2**/60.3|63.6/79.3/56.1|60.2/63.5/58.6|62.7
|1.0|**66.3**/71.8/**63.5**|**65.3**/81.6/**57.4**|**61.7/63.9/60.6**|**64.4**
|2.0|65.5/70.8/62.8|64.1/**83.0**/55.0|60.4/63.5/58.8|63.3


*[1] Saito, Kuniaki, and Kate Saenko. "Ovanet: One-vs-all network for universal domain adaptation." ICCV 2021.*

---

### Meta-Review · Area_Chair_Hk8K · 2024-12-15

**Metareview:**

This paper tackles the challenges of inherent biases and overlooked semantic distribution shifts in Generalized Category Discovery (GCD). It introduces a novel model incorporating a debiased auxiliary classifier, a semantic distribution detector, and a curriculum learning strategy to enhance the learning process. Experimental evaluations demonstrate that the proposed D2G model achieves state-of-the-art performance on GCD benchmarks. Reviewers provided positive feedback and expressed satisfaction with the authors' responses and revisions.

**Additional Comments On Reviewer Discussion:**

All the concerns are well addressed with an updated revision, especially, the newly added experimental results are very convincing, and more intuitive motivation about the problem setting.

---

### Decision · Program_Chairs · 2025-01-22

Accept (Poster)